# Glycosyl *ortho*-(1-phenylvinyl)benzoates versatile glycosyl donors for highly efficient synthesis of both *O*-glycosides and nucleosides

Penghua Li[1,2,3], Haiqing He [1,3], Yunqin Zhang[1,3], Rui Yang[1], Lili Xu[1], Zixi Chen[1], Yingying Huang[1], Limei Bao[1] & Guozhi Xiao [1]*

Both of *O*-glycosides and nucleosides are important biomolecules with crucial rules in numerous biological processes. Chemical synthesis is an efficient and scalable method to produce well-defined and pure carbohydrate-containing molecules for deciphering their functions and developing therapeutic agents. However, the development of glycosylation methods for efficient synthesis of both *O*-glycosides and nucleosides is one of the long-standing challenges in chemistry. Here, we report a highly efficient and versatile glycosylation method for efficient synthesis of both *O*-glycosides and nucleosides, which uses glycosyl *ortho*-(1-phenylvinyl)benzoates as donors. This glycosylation protocol enjoys the various features, including readily prepared and stable donors, cheap and readily available promoters, mild reaction conditions, good to excellent yields, and broad substrate scopes. In particular, the applications of the current glycosylation protocol are demonstrated by one-pot synthesis of several bioactive oligosaccharides and highly efficient synthesis of nucleosides drugs capecitabine, galocitabine and doxifluridine.

[1] State Key Laboratory of Phytochemistry and Plant Resources in West China, Kunming Institute of Botany, University of Chinese Academy of Sciences, Chinese Academy of Sciences, Kunming 650201, China. [2] School of Chemical Science and Technology, Yunnan University, Kunming 650091, China. [3]These authors contributed equally: Penghua Li, Haiqing He, Yunqin Zhang. *email: xiaoguozhi@mail.kib.ac.cn

Carbohydrates are essential molecules with important rules in numerous biological processes such as cell growth and proliferation, immune response, and viral and bacterial infection[1,2]. Due to the highly heterogeneous character of carbohydrate structures, it is a formidable task to isolate homogeneous and pure glycans from nature. Chemical synthesis is an efficient, reliable, and scalable means to produce well-defined glycans for deciphering their functions and developing pharmaceuticals[3–7]. During the past century, a variety of glycosylation methods have been developed to streamline the chemical synthesis of glycans[8–17]. In comparison with synthesis of O-glycosides, methods capable of efficient synthesis of nucleosides (N-glycosides) are relatively limited. Nucleosides are also important molecules, which played such crucial roles in many cellular processes as enzyme metabolism and regulation, cell signaling, and DNA and RNA synthesis[18]. Nucleosides natural products and nucleosides analogs usually demonstrate potent anticancer, antiviral, and antibacterial activities[19]. The development of the efficient methods for the synthesis of nucleosides still remains a high priority to develop therapeutic agents such as anticancer and antiviral drugs, to explore DNA sequencing technique, and to understand their mechanisms of action[20–22]. Most of common O-glycosylation donors including glycosyl trichloroacetimidates[23], trifluoroacetimidates[24], chlorides/bromides[25], phosphites[26], thioglycosides[27], sulfoxides[28], sugar 1,2-anhydrides[29], and n-pentenyl glycosides[30] have the only limited success for nucleosides synthesis (Fig. 1a). The problems inherent to nucleosides synthesis mainly include the poor solubility and nucleophilicity of nucleobases, which results in the unfavorable competition for glycosylation with other nucleophilic species derived from the leaving group and promoters in the coupling reaction. Therefore, the development of glycosylation methods for highly efficient synthesis of both O-glycosides and nucleosides is highly desirable and one of the long-standing challenges in chemistry.

A Vorbrüggen type reaction[31–33] is a famous method for the synthesis of nucleosides, which involves the glycosylation of sugar acetates with trimethylsilylated nucleobases under strong Lewis acid activation (Fig. 1b). However, due to the stoichiometric use of strong Lewis acids, e.g. trimethylsilyl triflate and SnCl$_4$, the limitations of this method are obvious, including: (1) the poor functional group compatibility of the strong Lewis acid; (2) the low coupling yields and moderate N9/N7 regioselectivity when purines are used; and (3) the issues of work-up and disposal, especially on preparative scale. To overcome these issues, Jamison's group[34] introduced pyridinium triflate salts as efficient bronsted acid catalysts for nucleosides and nucleosides analogs synthesis (Fig. 1b). Despite the success in nucleosides synthesis, sugar acetates are relatively poor donors for synthesis of O-glycosides. In 2008, Yu's group[35] introduced a glycosylation method with glycosyl ortho-alkynylbenzoates (ABz) as donors and gold(I) complexes as catalysts. This Yu glycosylation enjoys mild glycosylation reaction conditions and has been successfully demonstrated in the total synthesis for complex O-glycosides natural products and nucleosides antibiotics[10,36] (Fig. 1c). In 2019, the group of Yu[37] reported another highly effective and versatile glycosylation method with 3,5-dimethyl-4-(2′-phenylethynylphenyl)phenyl (EPP) glycosides as donors, which is suitable for efficient synthesis both of O-glycosides and nucleosides (Fig. 1d). Notwithstanding these breakthroughs, the challenges still remain in this realm. For example, methods using readily prepared and stable glycosyl donors, cheap and readily available promoters, and mild reaction conditions are still limited. The number of leaving groups that could be combined to achieve multi-steps one-pot synthesiss of oligosaccharides is still rare[13,14,38,39]. Furthermore, general glycosylation methods suitable for industrial appllications

with lower costs for efficient synthesis of both O-glycosides and nucleosides remain extremely rare.

Herein, we report a highly efficient and versatile glycosylation method with glycosyl ortho-(1-phenylvinyl)benzoates (PVB) as donors for efficient synthesis of both O-glycosides and nucleosides (Fig. 1e). Readily prepared and stable donors, cheap and readily available promoters, mild reaction conditions, good to excellent yields, and broad substrate scopes are highlighted in this glycosylation protocol. Furthermore, the applications of the current glycosylation protocol have been successfully demonstrated in the one-pot synthesis of several bioactive oligosaccharides and highly efficient synthesis of nucleosides drugs capecitabine, galocitabine, and doxifluridine (Fig. 1f).

## Results

**Conditions optimization.** We commenced with the preparation of glycosyl PVB donors, which were readily prepared by condensation of sugar hemiacetals with the ortho-(1-phenylvinyl) benzoic acid in the presence of EDCI, DMAP, and iPr$_2$EtN (Supplementary Methods). It was noteworthy that the ortho-(1-phenylvinyl)benzoic acid was easily obtained by one-step Wittig methenylation of the cheap 2-benzoylbenzoic acid (0.1$/g) in excellent yield[40]. The resultant glycosyl PVB donors (1a–t) are stable and stay inert on shelf for at least one month. Next, we investigated glycosylation of glucosyl perbenzoyl PVB 1a and 1,2:3,4-di-O-isopropylidene-α-galactoside 2a under various conditions (Table 1). A variety of promoters such as Ph$_3$PAuOTf, AgOTf, Cu(OTf)$_2$, TMSOTf and HOTf, and MeOTf were screened; however, glucosyl perbenzoyl PVB 1a stayed largely intact and no glycosylation reaction occurred, except the promoter HOTf affording the desired glycoside 3a in 9% yield. Fortunately, glycosylation of disarmed PVB 1a (1.2 equiv.) and galactoside 2a (1.0 equiv.) under the action of NIS (2.5 equiv.) and TMSOTf (0.3 equiv.) proceeded smoothly in CH$_2$Cl$_2$ at 0 °C to RT, providing the coupled (1 → 6)-disaccharide 3a in 97% yield. Reducing the amount of NIS from 2.5 equiv. to 1.5 equiv. still produced disaccharide 3a in an excellent yiled (94%). The departing species, isobenzofuran-1-one derivative 4 was readily isolated and confirmed unambiguously by X-ray diffraction analysis (CCDC 1939909) (Fig. 2).

**Scope of synthesis of O-glycosides.** With the optimal conditions for the model glycosylation reaction (1a + 2a → 3a) in hand, we began to investigate the scope of this glycosylation protocol for O-glycosides synthesis (Fig. 2). We were pleased to discover that a wide range of carbohydrate alcoholic acceptors including the sterically hindered C4 position of glucose and C2 position of mannose were coupled smoothly with 1a, providing β-(1 → 6)-, (1 → 4)-, (1 → 3)-, and (1 → 2)-disaccharides 3b–h in excellent yields (92–99%). It is worth noting that this glycosylation protocol works well for both of higher and lower nucleophilic carbohydrate alcoholic acceptors (e.g., 3c and 3d). A mild anesthetic menthol and hindered nucleophiles such as 1-adamantanol and benzyl oleanolate were also glycosylated uneventfully with 1a, producing the coupled glycosides 3i–k in 87–95% yields. Glycosylation of the acid and base-sensitive and electron-rich podophyllotoxin[41] with 1a also proceeded smoothly to furnish the desired podophylotoxin 4-O-glucoside 3l in 94% yield, which is the analog of the first-line antitumoral drugs etoposide and teniposide[42,43]. Furthermore, we investigated different glycosyl donors. For pyranosyl donors, a wide array of glycosyl PVB donors including D-galacto-, D-manno-, L-rhamno-, L-arabino-, L-fuco-, and D-xylopyranosyl donors were coupled successfully to provide the corresponding glycosides 3m–r in excellent yields (88–99%). Besides pyranosyl donors,

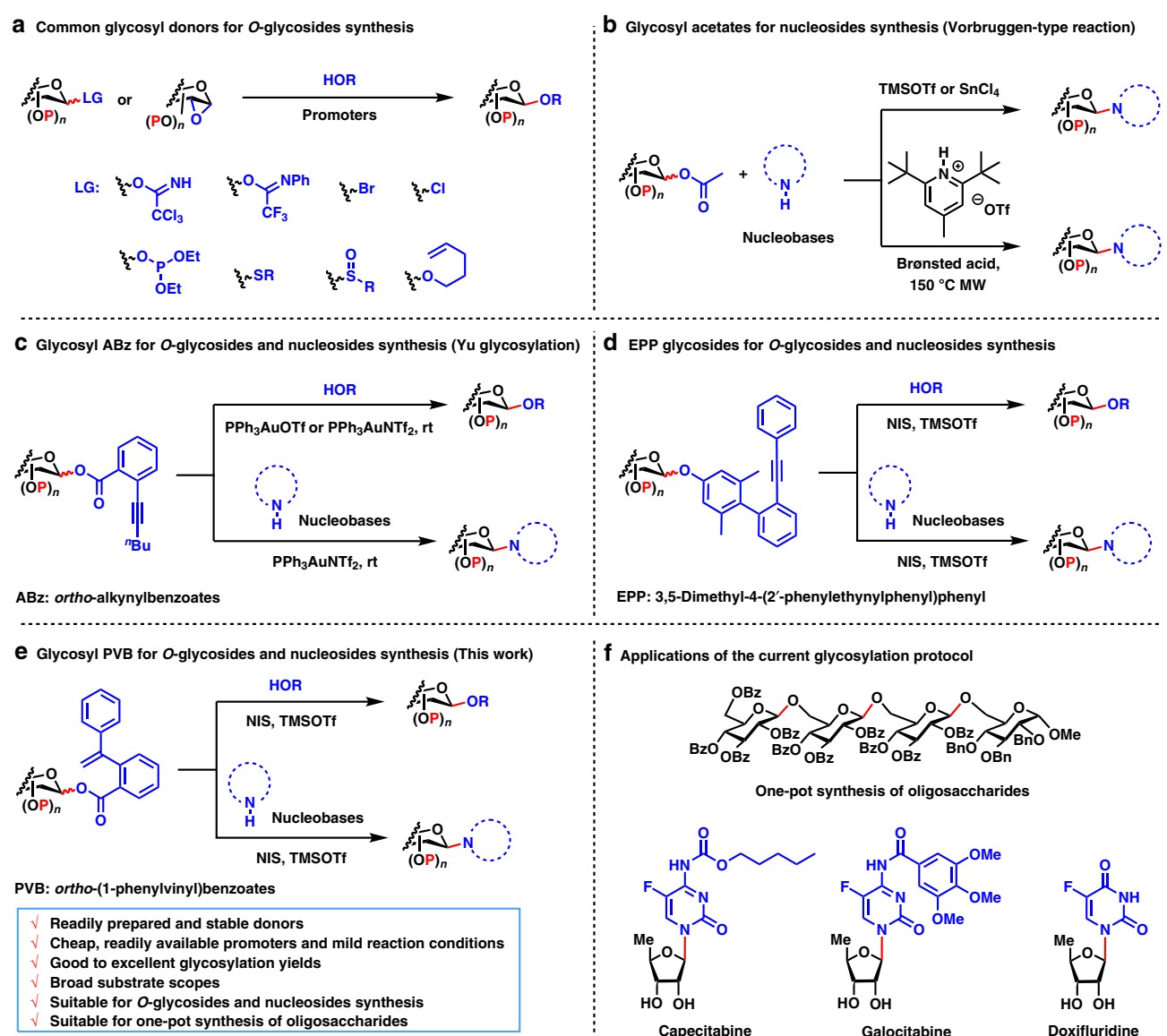

**Fig. 1 Glycosyl donors for synthesis of *O*-glycosides and nucleosides.** Prior art and current glycosylation protocol. Highlighted in blue are leaving groups and nucleophilic acceptors and in red are glycosidic linkages cleaved and formed. Bn benzyl, Bz benzoyl, NIS *N*-iodosuccinimide, TMSOTf trimethylsilyl trifluoromethanesulfonate, MW microwave.

furanosyl PVB donors including L-arabino-, D-ribo-, D-galacto-, and even D-fructofuranosyl donors were also glycosylated efficiently to furnish the corresponding glycosides **3s–3v** in nearly quantitative yields. It was noted that the famous Schmidt glycosylation method[44] using trichloroacetimidate (TCAI) as donors was not suitable for fructosylation due to the unsuccessful preparation of fructosyl trichloroacetimidate under various basic conditions[45]. The peracetyl gluco-, rhamno-, and xylopyranosyl PVB donors (**1b**, **1f** and **1h**) reacted uneventfully with secondary and primary sugar alcohols to give the corresponding glycosides **3w–3y** in 96–99% yield. It is noteworthy that when the corresponding glucosyl ABz was used as a donor, the complex mixture was obtained due to the production of orthoester-derived byproducts[46]. For both superarmed and armed glycosyl PVB donors[47], while **3z–3ab** were produced as the only β-products in 87–99% yields, an anomeric mixture of the glycoside **3ac** was isolated expectedly (99%, α/β = 1:1). For the formation of the challenging 2-deoxy glycosidic linkages, glycosyl PVB donor **1s** was also a viable substrate, and **3ad** was produced efficiently in 86% yield.

**N-glycosylation of pyrimidine nucleobases**. After the glycosyl PVB glycosylation protocol for efficient synthesis of *O*-glycosides was successfully set up, the N-glycosylation of pyrimidine nucleobases with glycosyl PVB was then investigated (Fig. 3). Treatment of the poorly soluble uridine **5a** (2.0 equiv.) with *N,O*-bis(trimethylsilyl)trifluoroacetamide (BSTFA) in acetonitrile produced the soluble silylated uridine. We found that it was necessary to heat the nucleobases to 50 °C in 30 min to insure the full solubility of nucleobases. The resulting soluble silylated uridine was coupled with the perbenzoyl ribofuranosyl PVB **1l** in the activation of NIS (1.5 equiv.) and TMSOTf (0.5 equiv.) at 0 °C to room temperature. To our good fortune, the reaction proceeded smoothly, providing the desired nucleoside **6a** cleanly in an excellent yield (93%). With the optimal condition for the N-glycosylation of pyrimidine nucleobases in hand, we began to investigate the scope of this glycosylation protocol. We were pleased to discover that a wide range of nucleobases including uridine **5a**, thymine **5b**, *N*4-benzoylcytosine **5c**, 5-fluorouridine **5d**, and trifluorothymine **5e** were glycosylated smoothly with ribofuranosyl donors **1l** and **1t**, affording the desired nucleosides

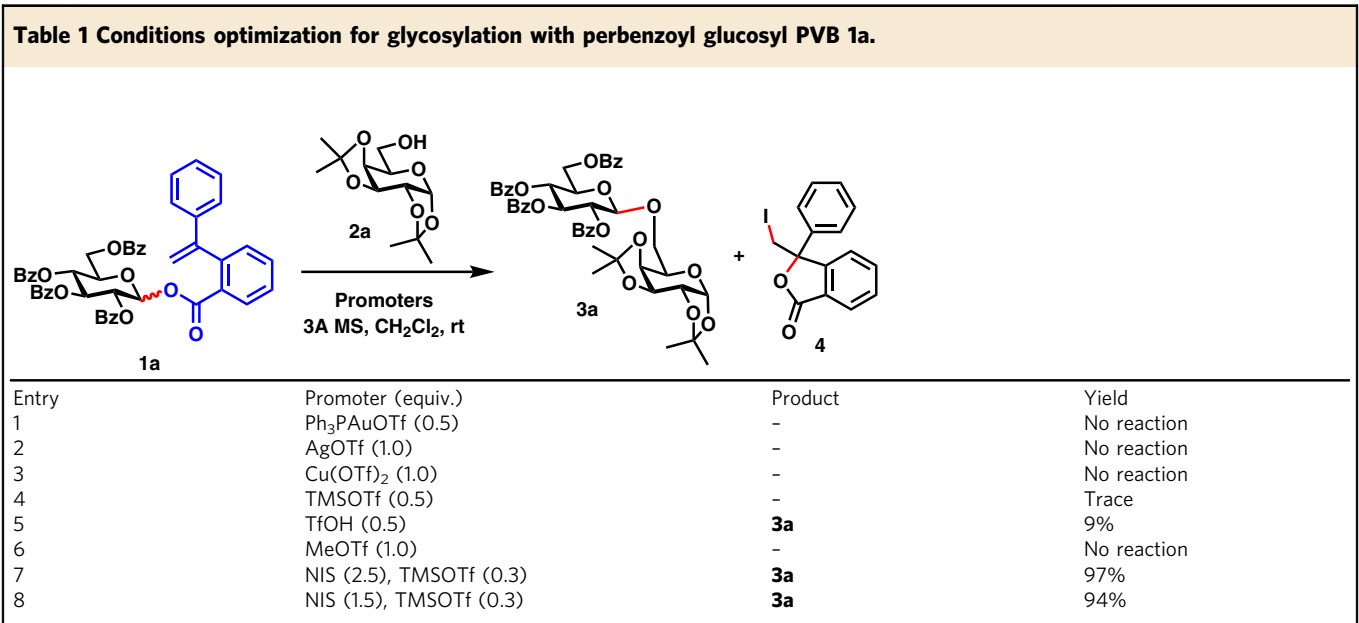

**Table 1 Conditions optimization for glycosylation with *perbenzoyl* glucosyl PVB 1a.**

| Entry | Promoter (equiv.) | Product | Yield |
|---|---|---|---|
| 1 | Ph₃PAuOTf (0.5) | – | No reaction |
| 2 | AgOTf (1.0) | – | No reaction |
| 3 | Cu(OTf)₂ (1.0) | – | No reaction |
| 4 | TMSOTf (0.5) | – | Trace |
| 5 | TfOH (0.5) | **3a** | 9% |
| 6 | MeOTf (1.0) | – | No reaction |
| 7 | NIS (2.5), TMSOTf (0.3) | **3a** | 97% |
| 8 | NIS (1.5), TMSOTf (0.3) | **3a** | 94% |

**6a–e** in excellent yields (85–96%). In comparison, using Yu glycosylation with glycosyl ABz, nucleosides **6a**, **6b**, and **6c** were obtained in 85%, 88%, and 95% yields, respectively[36]. Using EPP glycosides, **6a** and **6c** were produced in 90% and 91% yields, respectively[37]. The other furanosyl donors **1k** and **1m–n** including L-arabino, D-galacto-, and even D-fructofuranosyl donors were also coupled efficiently with nucleobases **5a**, **5c**, and **5e**, providing the corresponding nucleosides **6f–h** in excellent yields (91–96%). Besides the furanosyl donors, the pyranosyl PVB donors **1a**, **1e**, **1g**, and **1i** including D-gluco-, L-rhamno-, L-arabino-, and D-xylopyranosyl donors were also glycosylated smoothly with 5-fluorouridine **5d** and trifluorothymine **5e**, furnishing the corresponding nucleosides **6i–n** in excellent yields (89–99%). Interestingly, the big value (9.4 Hz) of the $J_{1,2}$ of nucleoside **6j** suggests a α-L-rhamnopyranose in ${}^4C_1$ conformation with the diaxial orientation of $H_1$ and $H_2$. It is noted that the fluoro and trifluoromethyl groups are intriguing structural motifs and frequently found in many pharmaceuticals. Thus, the incorporation of fluoro and trifluoromethyl groups into nucleosides may have great potential for the development of therapeutic agents[48,49].

**N-glycosylation of purines nucleobases**. We then examined the N-glycosylation of purines nucleobases with glycosyl PVB, which was a challenging task due to the N9/N7 regioselectivity issues of purines in the glycosylation reactions[50,51]. We chose two purine nucleobases, namely, *N*6-bis(*tert*-butoxycarbonyl)adenine **5f** and *N*2-*tert*-butoxycarbonyl-2-amino-6-iodopurine **5g**, which have already been efficiently utilized in Yu glycosylation of purines[36] and Mitsunobu-type N-alkylation reactions[52] with high N9/N7 regioselectivity. *Tert*-butoxycarbonyl (Boc) group installed in purines **5f** and **5g** solved the solubility issues of purine nucleobases and enabled the glycosylation to be conducted in dichloromethane. Boc-protected purine **5f** (1.0 equiv.) was glycosylated with the perbenzoyl ribofuranosyl PVB **1l** (1.2 equiv.) in the promotion of NIS (1.5 equiv.) and TMSOTf (0.3 equiv.) in dichloromethane at −20 °C. To our delight, the reaction proceeded efficiently, furnishing the desired N9 nucleoside **6o** in an excellent yield (84%). Coupling of *N*2-Boc-2-amino-6-iodopurine **5g** with PVB **1l** also led to the corresponding N9 6-iodopurine nucleoside **6p** in 85% yield, which is a useful precursor to the corresponding guanine nucleosides or 6-substituted analogs[52,53].

With the optimal condition for N-glycosylation of purines in hand, we went on to investigate the scope of this glycosylation protocol (Fig. 3). We were delighted to find that a wide array of furanosyl donors, including L-arabino-, D-5-deoxyribo-, D-galacto-, and even D-fructofuranosyl donors, were coupled smoothly, providing the desired N9 nucleosides **6q–t** in excellent yields (82–85%). Besides furanosyl donors, the pyranosyl donors including D-gluco-, D-galacto-, D-manno-, L-arabino-, D-xylo-, and L-rhamnopyranosyl donors also led to the coupled N9 nucleoside products **6u–z** in satisfactory yields (66–76%). Interestingly, nucleoside product **6w** bears a α-D-mannopyranose in ${}^1C_4$ conformation[54–56]. It was noted that N-glycosylation of purines using Yu glycosylation with glycosyl *ortho*-alkynylbenzoates afforded N9 nucleosides **6o**, **6p**, **6q**, and **6u** in 77%, 84%, 80%, and 48% yield, respectively[36]. Using EPP glycosides, the corresponding **6o**, **6q**, and **6u** were prepared in 74%, 73%, and 69% yields, respectively[37].

**Comparison of the donor reactivities**. Of note, the present glycosyl PVB activation conditions (NIS/TMSOTf) have been extensively utilized in thioglycosides and 4-*n*-pentenyl glycosides glycosylation reactions[57,58]. Therefore, the donor reactivities of glycosyl PVB with thioglycosides and 4-*n*-pentenyl glycosides were compared (Fig. 4). NIS (1.0 equiv.) and TMSOTf (0.3 equiv.) were added to a mixture of *p*-tolylthio perbenzoyl glucopyranoside (**7a**, 1.0 equiv.), perbenzoyl glucopyranosyl PVB (**1a**, 1.0 equiv.), and sugar alcohol (**2a**, 1.0 equiv.) in CH₂Cl₂ at −15 °C. Interestingly, after 2 h, *p*-tolylthio perbenzoyl glucopyranoside **7a** was fully recovered (98%), while perbenzoyl glucopyranosyl PVB **1a** was completely glycosylated with **2a**, providing the coupled disaccharide **3a** and isobenzofuran-1-one derivative **4** in 94% and 99% isolated yield, respectively. Furthermore, we compared the donor reactivity of disarmed PVB donor **1a** with armed perbenzyl glucothioglycoside, which shows about 2000 times reactivity of its perbenzoyl counterpart **7a**[59]. To our surprise, disarmed PVB donor **1a** was still more reactive and about three times reactivity of perbenzyl glucothioglycoside **7b** (Supplementary Fig. 7). In a similar competition reaction, while **1a** was fully coupled with **2a**, producing **3a** and **4** in a nearly quantitatively yield, perbenzoyl glucosyl *n*-Pen **7c** was recovered completely (99%). Furthermore, **1a** was about seven times more reactive than perbenzyl glucosyl *n*-Pen **7d** (Supplementary Fig. 9).

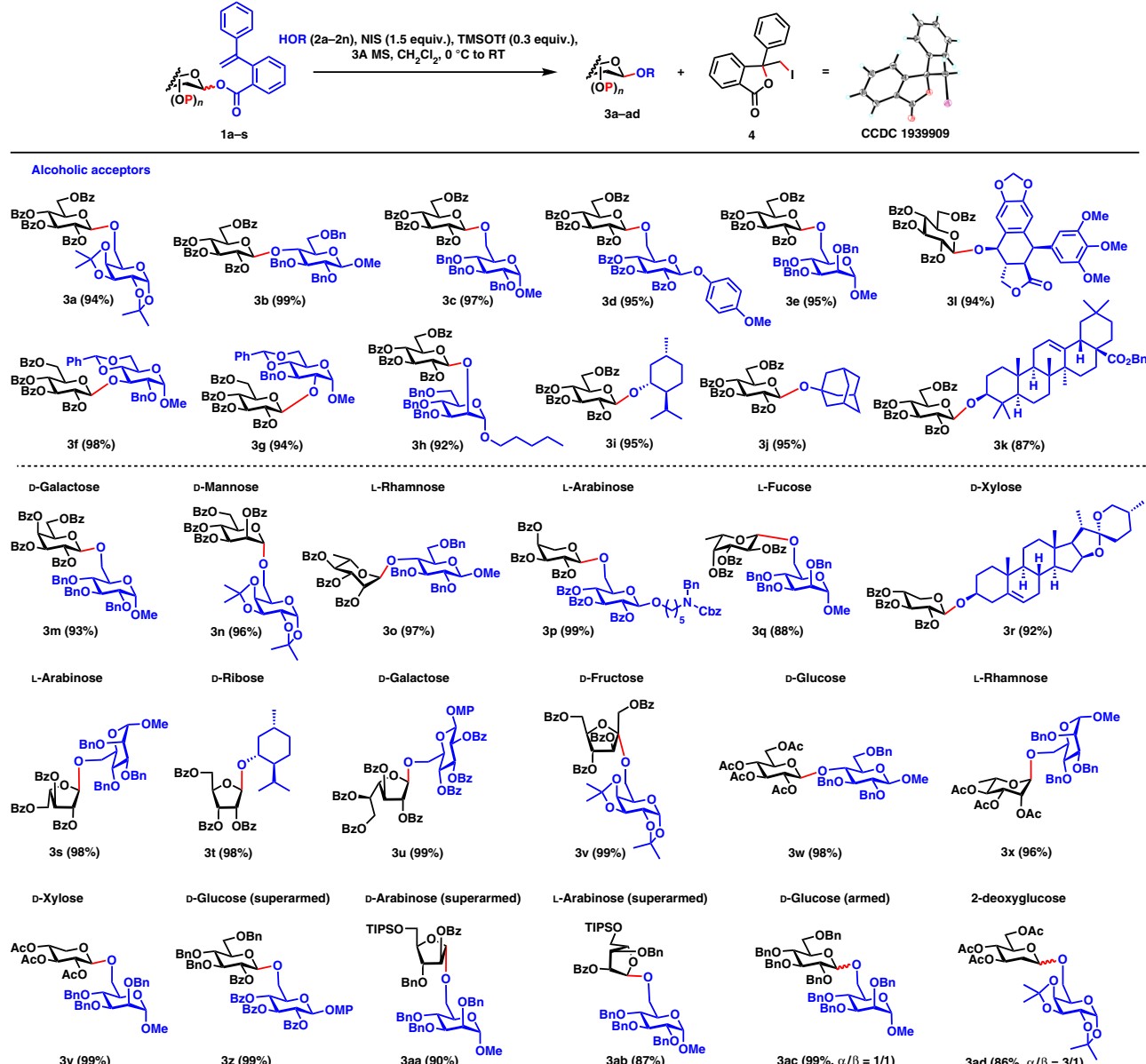

**Fig. 2 Scope of synthesis of *O*-glycosides with glycosyl PVB donors.** Highlighted in blue are leaving groups and nucleophilic acceptors and in red are chemical bonds cleaved and formed. Ac acetyl, TIPS triisopropylsilyl, MP 4-methoxyphenyl, Cbz benzyloxycarbonyl, MS molecular sieves, RT room temperature.

**One-pot synthesis of oligosaccharides 13–16.** We envisioned that large enough donor reactivity differences of glycosyl PVB donors with thioglycosides and 4-*n*-pentenyl glycosides could lead to one-pot synthesis of oligosaccharides. Indeed, for glycosyl PVB and thioglycoside pair, glycosylation of disarmed PVB donor **1a** (1.0 equiv.) with disarmed thioglycoside **8** (1.0 equiv.) under the activation of NIS (1.0 equiv.), and TMSOTf (0.3 equiv.) produced the coupled disaccharide intermediate, which was further coupled with the acceptor **2e** (1.0 equiv.) under the promotion of NIS (1.5 equiv.) and TMSOTf (0.3 equiv.), providing the trisaccharide **13** in 89% yield in the same pot (Fig. 4). For glycosyl PVB and *n*-Pen glycoside pair, PVB **1a** (1.0 equiv.) was coupled with *n*-Pen **9** (1.2 equiv.) under the promotion of NIS and TMSOTf to generate the coupled disaccharide intermediate, which was further glycosylated with the acceptor **2c** (1.1 equiv.) under the action of NIS and TMSOTf, providing trisaccharide **14** in 73% yield in a one-pot manner. Orthogonal one-pot synthesis of oligosaccharides is one of the most popular one-pot

glycosylation strategies[13,14]. However, the number of leaving groups that could be utilized in multistep orthogonal one-pot synthesis is still limited. Recently, the group of Xiao[38] developed orthogonal one-pot synthesis of oligosaccharides based on glycosyl ABz, which solved the shortcomings including aglycon transfer, high electrophilic character of departing species, and unpleasant odor inherent to thioglycosides based orthogonal one-pot glycosylation. We envisioned that orthogonal one-pot synthesis of oligosaccharides based on glycosyl PVB could also be explored to streamline chemical synthesis of oligosaccharides. Indeed, for glycosyl TCAI, ABz, and PVB triple, coupling of glucosyl TCAI **10** (1.2 equiv.) with ABz **11** (1.1 equiv.) under the catalysis of TMSOTf afforded disaccharide intermediate, which was further glycosylated with PVB **12** (1.0 equiv.) under the catalytic amount of Ph₃PAuOTf, providing the trisaccharide intermediate. The above trisaccharide intermediate was further glycosylated with the acceptor **2c** (0.9 equiv.) under the promotion of NIS and TMSOTf to give tetrasaccharide **15** in 74% yield

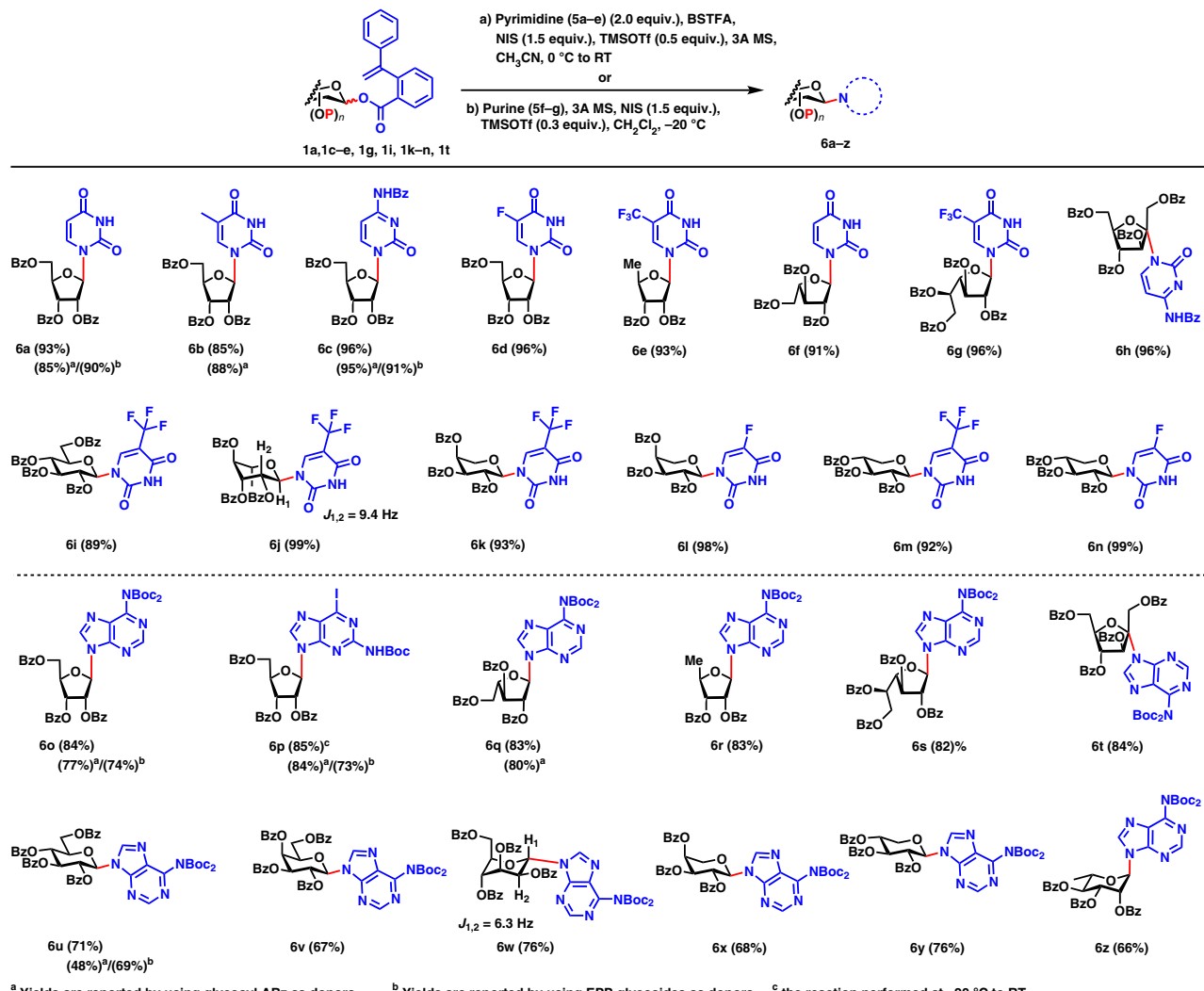

a) Pyrimidine (5a–e) (2.0 equiv.), BSTFA, NIS (1.5 equiv.), TMSOTf (0.5 equiv.), 3A MS, CH₃CN, 0 °C to RT

or

b) Purine (5f–g), 3A MS, NIS (1.5 equiv.), TMSOTf (0.3 equiv.), CH₂Cl₂, –20 °C

1a,1c–e, 1g, 1i, 1k–n, 1t → 6a–z

6a (93%) (85%)ᵃ/(90%)ᵇ · 6b (85%) (88%)ᵃ · 6c (96%) (95%)ᵃ/(91%)ᵇ · 6d (96%) · 6e (93%) · 6f (91%) · 6g (96%) · 6h (96%)

6i (89%) · 6j (99%) J₁,₂ = 9.4 Hz · 6k (93%) · 6l (98%) · 6m (92%) · 6n (99%)

6o (84%) (77%)ᵃ/(74%)ᵇ · 6p (85%)ᶜ (84%)ᵃ/(73%)ᵇ · 6q (83%) (80%)ᵃ · 6r (83%) · 6s (82%) · 6t (84%)

6u (71%) (48%)ᵃ/(69%)ᵇ · 6v (67%) · 6w (76%) J₁,₂ = 6.3 Hz · 6x (68%) · 6y (76%) · 6z (66%)

ᵃ Yields are reported by using glycosyl ABz as donors  ᵇ Yields are reported by using EPP glycosides as donors  ᶜ the reaction performed at –20 °C to RT

**Fig. 3 Scope of synthesis of nucleosides with glycosyl PVB donors.** Highlighted in blue are leaving groups and nucleophilic acceptors and in red are glycosidic bonds cleaved and formed. Boc *t*-butoxycarbonyl.

in one-pot, whose sulfated derivatives exhibit significant proangiogenic activity[60]. It was noted that stepwise orthogonal glycosylation approach afforded tetrasaccharide **15** in only 39% overall yield[61]. Replacing the above acceptor **2c** with the poor 4-OH acceptor **2b**, tetrasacchaide **16** was still obtained in satisfactory 59% overall yield in one pot by successive coupling of TACI **10**, ABz **11**, PVB **12**, and acceptor **2b** (Fig. 4).

**Synthesis of nucleosides drugs.** Finally, the utilities of this glycosylation protocol have been successfully demonstrated in the application to the efficient synthesis of nucleoside antibiotics capectibine, galocitabine, and doxifluridine (Fig. 5). In particular, capecitabine **17** is an important drug against breast and colorectal cancers[62–65] and is commercially available in the market under the brand name XELODA®. Using the above standard conditions for N-glycosylation with pyrimidines, pyrimidine **5h** was coupled efficiently with perbenzoyl 5-deoxyribofuranosyl PVB **1t**, providing the pyrimidine nucleoside in excellent yield (91%), which was treated with NaOH in MeOH/H₂O to produce nucleoside capecitabine **17** in 96% yield. It was noted that N-glycosylation of sugar acetate with pyrimidine **7 h** under the action of pyridinium triflate salts at 140 °C developed by the group of Jamison[34] resulted in significant decomposition of carbamate functionality and moderate glycosylation yield[66]. This problem rendered them

to first couple more electron rich and less reactive pyrimidine, then install the carbamate functionality, finally deprotect the acyl groups to achieve the synthesis of capectitabine **17** in 72% overall yield over three steps[66]. In comparsion, the current two-step synthesis of anticancer drug capecitabine **17** in 87% overall yield is clearly more efficient and less steps than any other previously reported processes[67–73]. The current glycosylation protocol highlighted the mild reaction condition, which is compatible with the carbamate functionality. In the similar glycosylation protocol, coupling of pyrimidines **5d** and **5i** with PVB **1t** afforded the corresponding pyrimidine nucleosides in excellent yields (83–98%), which was deprotected with NaOH in MeOH/H₂O to give doxifluridine **18** and galocitabine **19** in 91% and 95% yields, respectively.

**A plausible mechanism.** After the glycosyl PVB glycosylation protocols for highly efficient synthesis both of O-glycosides and nucleosides were successfully established and the synthetic applications toward one-pot synthesis of bioactive oligosaccharides and efficient synthesis of nucleosides antibiotics were successfully demonstrated, a plausible mechanism of the NIS/TMSOTf mediated glycosylation with glycosyl PVB as donors was outlined in Fig. 6. Activation of NIS with TMSOTf leads to the formation of iodonium species, which can attack a C–C double

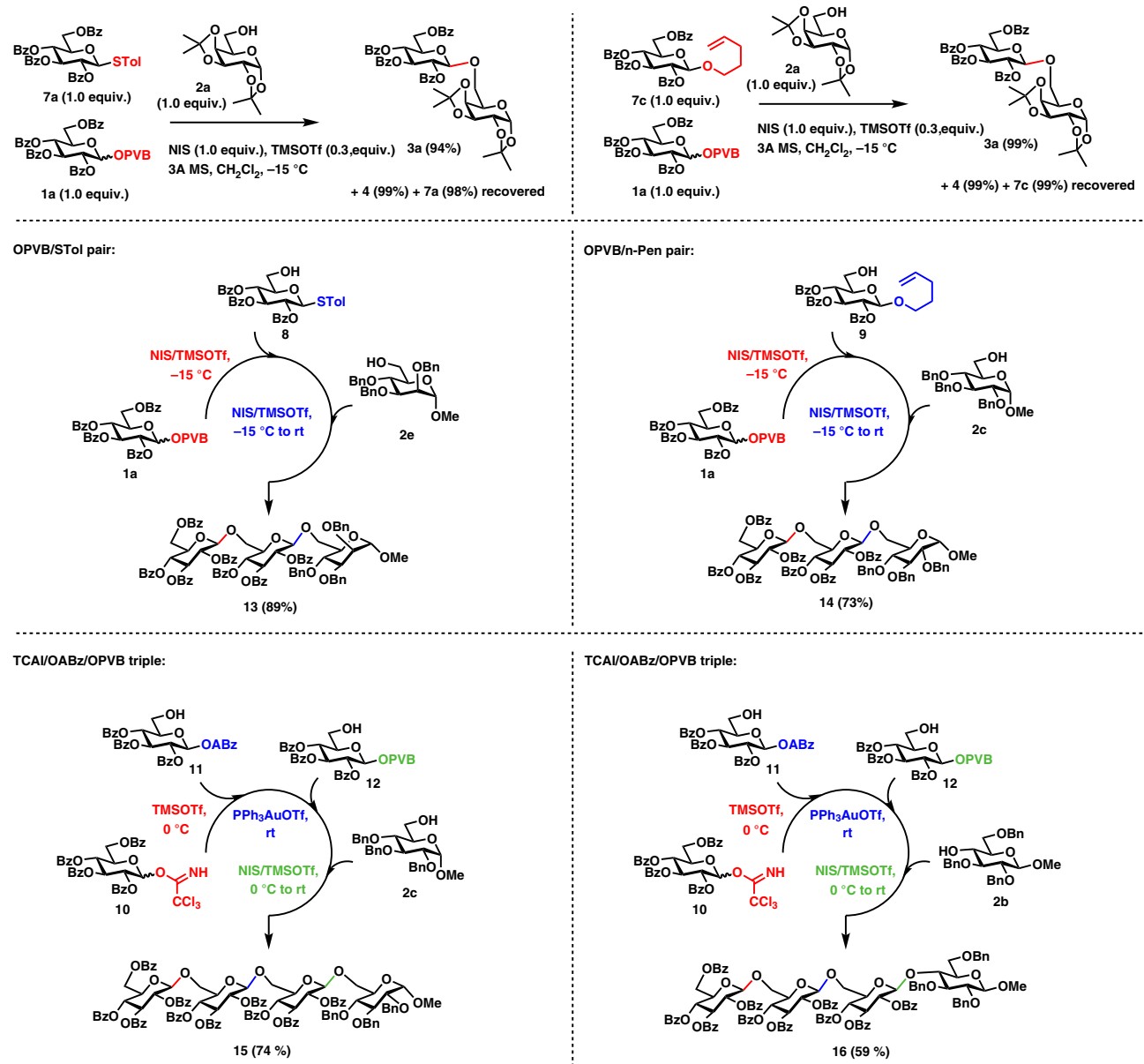

**Fig. 4 Comparison of the donor reactivities and one-pot synthesis of oligosaccharides 13–16.** Highlighted in red, blue, and green are the corresponding leaving groups, activation conditions, and glycosidic bonds formed. Tol *p*-tolyl, ABz *ortho*-alkynylbenzoates, PVB *ortho*-(1-phenylvinyl)benzoates, Pen pentenyl, TCAI trichloroacetimidates.

bond within the aglycone of glycosyl PVB **1**, resulting in the generation of diphenylmethyl cation species **A**. We envisioned that this cation species **A** can be stabilized by two adjacent phenyl groups, facilitating the ready cleavage of the anomeric C–O bond to generate the glycosyl oxocarbenium species **B** and the departing species of the leaving group, isobenzofuran-1-one derivative **4**. The oxocarbenium **B** can be attacked by alcoholic nucleophiles **2** or nucleobases **5** to produce *O*-glycosides products **3** or nucleosides **6** and H⁺. Protonation of silylated succinimide regenerates TMSOTf, which can undergo the next catalytic cycle.

## Discussion
We have developed a highly efficient and versatile glycosylation method for the highly efficient synthesis of both *O*-glycosides and nucleosides, which uses glycosyl PVB as donors and NIS/TMSOTf as promoters. This glycosylation protocol highlighted the various advantages, including (1) readily prepared and stable glycosyl PVB

donors; (2) cheap and readily available promoters, and mild reaction conditions; (3) good to excellent glycosylation yields; and (4) broad substrate scopes. Furthermore, one-pot synthesis of oligosaccharides based on glycosyl PVB highly streamlined chemical synthesis of oligosaccharides, which avoided three major issues including aglycon transfer, undesired interference of departing species, and unpleasant odor inherent to thioglycosides based one-pot glycosylation. Highly efficient synthesis of nucleosides drugs capecitabine, galocitabine, and doxifluridine using the current glycosylaton protocol has also been successfully demonstrated. With these exciting features, we believe this glycosylation protocol will find broad applications in chemical synthesis and inspire the development of other glycosylation methods for efficient synthesis of both *O*-glycosides and nucleosides.

## Methods
**General**. The complete experimental details, compound characterization data, and NMR spectra of compounds synthesized in this study see Supplementary information.

**Fig. 5 Application to the synthesis of nucleosides Capecitabine 17, Doxifluridine 18, and Galocitabine 19.** Highlighted in blue are leaving groups and nucleophilic acceptors and in red are glycosidic bonds formed.

**Fig. 6 A plausible mechanism of the glycosyl PVB glycosylation protocol.** Highlighted in red are active species and chemical bonds cleaved and formed; highlighted in blue are leaving groups and nucleophilic acceptors and in light blue are oxocarbenium intermediates.

**General procedure for glycosylation with alcoholic acceptors.** A solution of glycosyl PVB donor **1** (1.2 equiv.) and alcoholic acceptors **2** (1.0 equiv.) in dry CH$_2$Cl$_2$ (0.033 M) was stirred at room temperature for 30 min in the presence of activated 3 Å MS (3.0 g/mmol) under argon atmosphere. Then the vessel was chilled to 0 °C, to which NIS (1.5 equiv.) and TMSOTf (0.3 equiv.) were added. The reaction mixture was stirred for 2 h after the temperature gradually rose to room temperature. Then Et$_3$N was added to quench the reaction and the solvent was removed under reduced pressure. The resulting residue was purified by silica gel column chromatography to afford the glycosylated product.

**General procedure for glycosylation with pyrimidines.** BSTFA (4.0 eq) was added to a stirred suspension of pyrimidine acceptors **5a–e** and **5h–i** (2.0 equiv.) in dry CH$_3$CN (0.1 M) under argon atmosphere. After the mixture was stirred at 50 °C for 30 min, this solution was added to a solution of glycosyl PVB donor **1** (1.0 eq) and activated 3 Å MS (4.0 g/mmol) in dry CH$_3$CN (0.05 M), which has been stirred at room temperature for 30 min under argon atmosphere. The stirring was continued for 10 min, then the vessel was chilled to 0 °C, to which NIS (1.5 equiv.) and TMSOTf (0.5 equiv.) were added. The reaction mixture was stirred for

3 h after the temperature gradually rose to room temperature. Et$_3$N was added to quench the reaction and the solvent was removed under reduced pressure. The resulting residue was purified by silica gel column chromatography to afford the glycosylated product.

**General procedure for glycosylation with purines.** A solution of glycosyl PVB donor **1** (1.2 equiv.) and purines acceptors **5f–g** (1.0 equiv.) in dry CH$_2$Cl$_2$ (0.033 M) was stirred at room temperature for 30 min in the presence of activated 3 Å MS (3.0 g/mmol) under argon atmosphere. Then the vessel was chilled to −20 °C, to which NIS (1.5 equiv.) and TMSOTf (0.3 equiv.) were added. The reaction mixture was stirred for 2 h after the temperature gradually rose to room temperature. Then Et$_3$N was added to quench the reaction and the solvent was removed under reduced pressure. The resulting residue was purified by silica gel column chromatography to afford the glycosylated product.

## Data availability

The authors declare that the data supporting the findings of this study are available within the article and its Supplementary Information Files. The X-ray crystallographic

coordinates for compound **4** reported in this study have been deposited at the Cambridge Crystallographic Data Centre (CCDC), under deposition numbers CCDC 1939909. These data can be obtained free of charge from The Cambridge Crystallographic Data Centre via www.ccdc.cam.ac.uk/data_request/cif.

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

## Acknowledgements

Financial support from the CAS Pioneer Hundred Talents Program (No. 2017-128) and the Start-up funding of Kunming Institute of Botany is greatly acknowledged. We also thank Dr. Xiaonian Li (Kunming Institute of Botany) for single crystal X-ray diffraction analysis.

## Author contributions

G.X. conceived the idea, designed the research, and wrote the manuscript with feedback from all authors. P.L. made the initial discovery. P.L., H.H. and Y.Z. explored the scope of this method. P.L. and H.H. applied this method to one-pot synthesis of oligosaccharides. H.H. applied this method to synthesis of nucleosides antibiotics. R.Y., L.X., Z.C. Y.H. and L.B. conducted the large scale preparation of the starting materials.

## Competing interests

The authors declare no competing interests.
