## [Peer Review File · Nature Communications]

Reviewers' comments:

Reviewer #1 (Remarks to the Author):

The authors describe the synthesis and application of Glycosyl ortho-(1-phenylvinyl)benzoates (PVB) as versatile glycosyl donors. These donors work well for oxygen and nitrogen nucleophiles, and can be selectively activated over other types of leaving groups. This is a carefully executed study that investigates all basic aspects of the new leaving group, including the mechanism of the activation, verified via the departed aglycone isolation and characterization (X-ray). The authors also investigate many pyranose and furanose sugar series, but somehow do not include aminosugars (not a criticism, just curious)

Advantages: broad substrate specificity, common activation conditions/promoters, relatively simple synthesis, high yields,

Disadvantages: designer leaving group that needs to be synthesized, anomeric mixtures of the donors, atom economy, stoichiometric NIS/TfOH activation.

Overall, this is a very interesting study that is very well executed, and showcase how new (designer) leaving groups should be introduced and studied. I support publication after the following points have been considered.

A few minor corrections

privileged donors (text and Schemes)? Do the authors mean common?

I am not a big fan of the term "glycosylation donors" adopted by the authors. There is a standard term : glycosyl donors that should be used instead

„sugar lactol derivatives“ is correct but „hemiacetals“ would be better

Statement: „The number of leaving groups that could be combined to achieve multi-steps one-pot synthesis of oligosaccharides is still rare.13-14,38-39“ can be supported by the following review on selective activation: Kaeothip, S. and A. V. Demchenko (2011). "Expeditious oligosaccharide synthesis via selective and orthogonal activation." Carbohydr. Res. 346: 1371-1388.

Under results the authors state "Next, we investigated glycosylation of disarmed glucosyl perbenzoyl..." the authors need to either introduce the concept of armed-disarmed with references or remove „disarmed,“ which is irrelevant in this context.

Under Scope, the authors state "regardless of the electron-donating or electron-withdrawing properties of carbohydrate alcoholic acceptors..." I understand that the authors actually mean "nucleophilicity of acceptors", but this statement is ultimately misleading. Please consider rephrasing.

During comparison of the donor reactivities study the authors use NIS (1.0 equiv) and TMSOTf (0.3 equiv), why do not they use the same conditions for "normal" glycosylations wherein 1.5 equiv NIS is used?

Reviewer #2 (Remarks to the Author):

The authors report a new glycosylation protocol using glycosyl ortho-(1-phenylvinyl)benzoates (PVB) as donors. This new type of donor is easily prepared, remains stable on shelf, and readily proceeds glycosylation under the action of NIS/TMSOTf. A broad substrate scope has been demonstrated, due to the mild reaction conditions. High yielding preparation of nucleosides by the present method, including the clinical drugs capecitabine, galocitabine and doxifluridine, is impressive. Moreover, the new method has been applied to the one-pot synthesis of glycans in combination with the use of conventional methods with trichloroacetimidates, ortho-alkynylbenzoates, and thioglycosides as donors. The experiments are well carried out and data well presented in the report.

Although many new glycosylation protocols have been disclosed in recent years, the present one shows real promise as a practically useful glycosylation method. Such a method requires easy preparation and activation of the donors and a wide substrate scope. Given the excellent performance of the new glycosylation method reported in the manuscript, I recommend its publication in Nat. Commun.

Minor points:

1) Page 95: the authors write "extremely hindered nucleophiles such as 1-adamantanol and benzyl oleanolate were also glycosylated". In fact, 1-adamantanol and benzyl oleanolate are hindered, but not extremely hindered nucleophiles.

2) Scheme 6 and the relevant context: the possible phenyl conjugated cations B and C are described as intermediates which are in equilibrium with cation A. Intermediates B and C are not important to the glycosylation reaction, nor their occurrence is proven, therefore are better to be removed.

Reviewer #1:

privileged donors (text and Schemes)? Do the authors mean common?

I am not a big fan of the term “glycosylation donors” adopted by the authors. There is a standard term : glycosyl donors that should be used instead

„sugar lactol derivatives “ is correct but „hemiacetals “ would be better

Response: We changed “privileged” to “common” in the revised manuscript. We changed “glycosylation donors” and “sugar lactol derivatives” to “glycosyl donors” and “sugar hemiacetals” respectively in the revised manuscript.

Statement: „The number of leaving groups that could be combined to achieve multi-steps one-pot synthesis of oligosaccharides is still rare.13-14,38-39 “ can be supported by the following review on selective activation: Kaeothip, S. and A. V. Demchenko (2011). "Expeditious oligosaccharide synthesis via selective and orthogonal activation." *Carbohydr. Res.* 346: 1371-1388.

Response: Indeed, we agree that the number of leaving groups that could be combined to achieve multi-steps one-pot synthesis of oligosaccharides is still rare.13-14,38-39 “ can be supported by the following review on selective activation: Kaeothip, S. and A. V. Demchenko (2011). "Expeditious oligosaccharide synthesis via selective and orthogonal activation." *Carbohydr. Res.* 346: 1371-1388. However, the recent review (Panza, M.; Pistorio, S. G.; Stine, K. J.; Demchenko, A. V. Automated Chemical Oligosaccharide Synthesis: Novel Approach to Traditional Challenges. *Chem. Rev.* 2018, **118**, 8105–8150) including the content of the reference (*Carbohydr. Res.* 346: 1371-1388) the reviewer suggested has been cited in the manuscript reference 14. Therefore, we don't think an addition to the reference list (*Carbohydr. Res.* 2011, 346, 1371-1388) is needed.

Under results the authors state “ Next, we investigated glycosylation of disarmed glucosyl perbenzoyl... “ the authors need to either introduce the concept of armed-disarmed with references or remove „disarmed, “ which is irrelevant in this context.

Response: We removed “disarmed” in the revised manuscript.

Under Scope, the authors state “ regardless of the electron-donating or electron-withdrawing properties of carbohydrate alcoholic acceptors...” I understand that the authors actually mean “nucleophilicity of acceptors” , but this statement is ultimately misleading. Please consider rephrasing.

Response: We changed “regardless of the electron-donating or electron-withdrawing

properties of carbohydrate alcoholic acceptors···” to “for both of higher and lower nucleophilic carbohydrate alcoholic acceptors” in the revised manuscript.

During comparison of the donor reactivities study the authors use NIS (1.0 equiv) and TMSOTf (0.3 equiv), why do not they use the same conditions for “normal” glycosylations wherein 1.5 equiv NIS is used?

Response: For “normal” glycosylations, 1.5 equiv NIS is used to insure the completeness of the glycosylation reactions. If 1.0 equiv NIS is used in the general glycosylation reaction conditions, some glycosylation reactions are not complete. During comparison of the donor reactivities, NIS (1.0 equiv) and TMSOTf (0.3 equiv) were used to avoid the activation of donors with excess of NIS. If 1.5 equiv NIS is used in the comparison of the donor reactivities, excess 0.5 equiv NIS can activate the less reactive donors, which results in the incorrectness of comparison studies.

Reviewer #2:

1) Page 95: the authors write “extremely hindered nucleophiles such as 1-adamantanol and benzyl oleanolate were also glycosylated” . In fact, 1-adamantanol and benzyl oleanolate are hindered, but not extremely hindered nucleophiles.

Response: We removed “extremely” in the revised manuscript.

2) Scheme 6 and the relevant context: the possible phenyl conjugated cations B and C are described as intermediates which are in equilibrium with cation A. Intermediates B and C are not important to the glycosylation reaction, nor their occurrence is proven, therefore are better to be removed.

Response: We removed “Intermediates B and C” in Scheme 6 and the relevant context in the revised manuscript.

We highlighted the changes in the revised manuscript.

We thank the above two anonymous reviewers for your support and recommendation!